# Structural Characterization of, and Protective Effects Against, CoCl_2_-Induced Hypoxia Injury to a Novel Neutral Polysaccharide from *Lycium barbarum* L.

**DOI:** 10.3390/foods14030339

**Published:** 2025-01-21

**Authors:** Yunchun Li, Jianfei Liu, Dong Pei, Duolong Di

**Affiliations:** CAS Key Laboratory of Chemistry of Northwestern Plant Resources and Key Laboratory for Natural Medicine of Gansu Province, Lanzhou Institute of Chemical Physics, Chinese Academy of Sciences, No. 18, Tianshui Middle Road, Lanzhou 730000, China; liyunchun@licp.cas.cn (Y.L.); jfliu@licp.cas.cn (J.L.); dongpei@licp.cas.cn (D.P.)

**Keywords:** *Lycium barbarum* polysaccharides, structural characterisation, oxidative stress, ischaemic stroke

## Abstract

Oxidative stress is closely related to the occurrence and development of ischaemic stroke. Natural plant polysaccharides have potential value in inhibiting oxidative stress and preventing ischaemic stroke. Here, a novel neutral polysaccharide named LICP009-3F-1a with a Mw of 10,780 Da was separated and purified from *Lycium barbarum* L. fruits. Linkage and NMR data revealed that LICP009-3F-1a has the following backbone: →4)-β-D-Glc*p*-(1→6)-β-D-Gal*p*-(1→, with a branched chain of β-D-Gal*p*-(1→3)-β-D-Gal*p*-(1→, α-L-Ara*f*-(1→ and →6)-α-D-Glc*p*-(1→ connected to the main chain through *O*-3 of →3,6)-β-D-Gal*p*-(1→. X-ray and SEM analyses showed that LICP009-3F-1a has a semicrystalline structure with a laminar morphology. Thermal property analysis showed that LICP009-3F-1a is thermally stable. In vivo experiments suggested that LICP009-3F-1a could inhibit hypoxia-induced oxidative stress damage by eliminating ROS, reversing and restoring the activities of the antioxidant enzymes SOD, CAT, and GPx, and reducing the expression levels of the HIF-1α and VEGF genes. Blocking the apoptosis genes Bax and Caspase 3 and upregulating the expression of the antiapoptotic gene Bcl-2 protected PC12 cells from hypoxia-induced apoptosis. These results suggest that LICP009-3F-1a may have multiple potential uses in the treatment of IS.

## 1. Introduction

Ischaemic stroke (IS) is one of the three major killer diseases that threaten human life and health [1,2]. Its pathogenesis is caused by cerebral hypoxia and the accumulation of reactive oxygen species caused by the interruption of local cerebral arterial blood supply, which induces irreversible damage to nerve cells, resulting in neurological deficits and even death [3,4,5,6]. Although the pathological mechanisms involved in IS are very complex, the instantaneous overproduction of reactive oxygen species (ROS) during the onset of IS will overwhelm the body’s antioxidant defence system and then cause a series of physiological events, such as inflammation, lipid peroxidation, blood–brain barrier damage, autophagy, and apoptosis, leading to neurodegeneration and apoptosis in neuronal cells [7,8,9,10]. Therefore, oxidative stress plays a major role in IS and is considered to be a key cause of death in stroke patients, and the inhibition of oxidative stress could become the main strategy in the prevention and treatment of IS [11,12].

Oxygen (O_2_) plays a crucial role in supporting cellular and organ function by enabling energy production through the mitochondrial respiratory chain. This process drives the proper operation of numerous enzymes and cellular activities [13]. Oxidative stress and apoptosis are the most important intracellular changes induced by hypoxia [14]. Hypoxia can increase the production of intracellular ROS. Excessive ROS attack membranes and proteins and lead to changes in the structure and function of DNA, resulting in cellular damage [15]. They also induce alterations in the inner mitochondrial membrane, leading to the opening of mitochondrial permeability transition (MPT) pores, disruption to the mitochondrial membrane potential, and the release of pro-apoptotic proteins from the intermembrane space into the cytosol. These changes ultimately lead to mitochondria-dependent apoptosis [16,17].

Many studies have shown that polysaccharide from *Lycii fructus* is an excellent antioxidant that can scavenge various free radicals, upregulate antioxidant enzyme activity, block lipid peroxidation, defend against oxidative stress damage, and activate antioxidative stress pathways, with other mechanisms exerting antioxidant effects, thereby showing various biological effects, such as the regulation of immunity, delays in ageing, antitumour effects, anti-Alzheimer’s disease effects, the protection of the nervous system, and the prevention and treatment of eye diseases [18,19,20,21]. And it has also been shown to inhibit apoptosis in cells exposed to oxidative stress, hypoxia, or other damaging conditions [22,23]. In this work, a neutral polysaccharide was isolated from *Lycii fructus* and shown to protect PC12 cells from CoCl_2_-induced hypoxic injury and apoptosis, which could prevent early IS.

## 2. Materials and Methods

### 2.1. Materials and Chemicals

*Lycii fructus* (Ningxia Zhongqi Wolfberry Trading Group Co., Ltd., Ningxia, China). Standard monosaccharide (Sigma–Aldrich, St. Louis, MO, USA). Sephadex G-50 (Solarbio Science & Technology Co., Ltd., Beijing, China)). All other reagents (if not mentioned) were of analytical grade or the highest grade available.

### 2.2. Preparation of LICP009-3F-1a

The isolation and purification of the neutral polysaccharide labelled “LICP009-3F-1” from *Lycii fructus* were performed as described in our previous procedure [24]. In brief, *L*. *barbarum* was subjected to high-speed shear-assisted extraction and cascade membrane separation with molecular weights of 10 and 5 kDa to obtain the brown crude polysaccharide LICP009, which was further purified through the process of protein removal, pigment removal, and ion exchange resin separation to give LICP009-3F-1. Then, the fraction containing LICP009-3F-1 was further separated by a Sephadex G-50 (2.6 cm × 30 cm) column to obtain the target polysaccharide LICP009-3F-1a.

### 2.3. Molecular Weight Analysis

The molecular weight (Mw) of LICP009-3F-1a was determined using high-performance size-exclusion chromatography (HPSEC) on an Agilent 1290 HPLC system (Agilent Technologies, Wilmington, DE, USA). The system was equipped with a multi-angle laser light scattering detector (DAWN HELEOS, Wyatt Technology Co., Santa Barbara, CA, USA) and a refractive index detector (RID-G7162B). Separation was achieved using two columns, OHpak SB-803 HQ (8.0 × 300 mm) and OHpak SB-804 HQ (8.0 × 300 mm), connected in series. The analysis was conducted at 30 °C with a flow rate of 0.5 mL/min and a sample injection volume of 100 μL.

### 2.4. Monosaccharide Composition Analysis

The analysis of monosaccharide composition was performed using ion chromatography (IC) (ICS500+, Thermo Fisher Scientific, Carlsbad, CA, USA) with equipped amperometric detection (PAD), following previously described methods. In brief, LICP009-3F-1a (5 mg) was hydrolysed with 4 mL 2 M Trifluoroacetic acid (TFA), followed by drying under a N_2_ stream. A 25 μL aliquot was injected into the IC system to be analysed [24].

### 2.5. FT-IR Spectroscopy

LICP009-3F-1a was mixed with KBr and ground with agate mortar. Then, infrared spectrum scanning analysis was carried out with a Thermo Nicolet Nexus 870 ESP FT-IR spectrometer (SpectraLab Scientific Inc., Markham, ON, Canada) at 4000–600 cm^−1^.

### 2.6. X-Ray Diffraction (XRD)

XRD (Siemens D 5000, Bruker, Karlsruhe, Germany) was used to determine the amorphous or crystalline structures of LICP009-3F-1a. Patterns were collected in the 2θ range of 5–80° with a step size of 0.02 and a speed of 5 s/step.

### 2.7. Thermogravimetric Analysis (TGA)

TGA for LICP009-3F-1a was performed on an STA449C thermogravimetric analyser (Mettler Toledo, Columbus, OH, USA) at 10 °C/min in the 35–800 °C temperature range.

### 2.8. Congo Red Test

Congo red was used to analyse the conformational structure of LICP009-3F-1a according to the method described in previous articles [24]. In brief, 2 mg of LICP009-3F-1a was added to 1 mL of deionized water and 1 mL of 80 μmol/L Congo red solution. Then, 0.5 M NaOH solution was added to the mixture to achieve a final concentration in the range of 0–0.5 mol/L. Deionized water was used as the control. The samples were scanned with a UV–vis spectrograph, and the maximum absorption wavelength was measured under NaOH. Distilled water was used as a blank control.

### 2.9. Scanning Electron Microscopy (SEM)

LICP009-3F-1a was placed onto an aluminium plate and sputter-coated with gold as described in our previous procedure. SEM images were obtained using a JSM 5600LV scanning electron microscope (Jeol Ltd., Tokyo, Japan) at ×500 and 2000 magnifications.

### 2.10. Linkage Pattern Analysis

LICP009-3F-1a was methylated using a previously described procedure with some minor modifications [25]. Briefly, 2 mL DMSO was added to 5 mg of LICP009-3F-1a, and anhydrous NaOH powder (50 mg) was added, followed by vigorous stirring. Then, CH_3_I (1 mL) was added dropwise while cooling in an ice bath, followed by sonication at 40 °C for 2 h in the dark. Then, 2 mL distilled water was used to quench the reaction, and excess CH_3_I was removed to obtain a methylated sample. The sample was treated with 2 M TFA for 1.5 h at 100 °C to further hydrolysis. The hydrolysate was reduced with NaBH_4_ and acetylated with pyridine and acetic anhydride. The final product was analysed by gas chromatography–mass spectrometry (GC–MS) using an Agilent 7890A-5977B system (Agilent Technologies Inc., Santa Clara, CA, USA). The temperature was maintained at 140 °C for 2 min and then increased to 230 °C at 3 °C/min over 20 min; 1.0 mL/min helium was used as the carrier gas. The injection volume was 1 μL, and the split ratio was 10:1.

### 2.11. Nuclear Magnetic Resonance (NMR) Analysis

LICP009-3F-1a (50 mg) was dissolved in 0.4 mL of D_2_O, freeze-dried, and re-dissolved in 0.4 mL of fresh D_2_O. This deuterium exchange process was repeated at least three times to ensure the complete replacement of H_2_O with D_2_O [26]. Subsequently, NMR, NMR, and two-dimensional spectra, including HSQC, COSY, HMBC, NOESY, and HHCOSY, were recorded using a Bruker AVANCE III NMR spectrometer operating at 600 MHz.

### 2.12. CoCl_2_-Induced Hypoxia Injury

PC12 cells (1 × 10^4^ cells/well) were cultured in 96-well plates and exposed to increasing concentrations of *Cobalt chloride* CoCl_2_ (0–900 µM) to simulate hypoxic conditions. Following 24 h of incubation in a humidified CO_2_ incubator at 37 °C, 10 µL of MTT solution (5 mg/mL) was added to each well, and the plates were further incubated for 2 h at 37 °C. The MTT solution was then removed, and 200 µL of DMSO was added to dissolve the formazan crystals. Absorbance was measured at 490 nm using a SpectraMax Plus plate reader (Molecular Devices, San Jose, CA, USA).

### 2.13. Cell Proliferation Assay Following LICP009-3F-1a Treatment

PC12 cells (1 × 10^4^ cells/well) were plated in 96-well plates and treated with varying concentrations of LICP009-3F-1a (0–500 μg/mL). Following a 24 h pretreatment in a humidified CO_2_ incubator at 37 °C, the cells were exposed to 400 μM CoCl_2_ to induce hypoxia. After 24 h of incubation, 10 µL of MTT solution (5 mg/mL) was added to each well, and the plates were incubated for an additional 2 h at 37 °C. MTT was removed, and an equal volume of 200 µL DMSO was added, after which the absorbance was measured at 490 nm.

### 2.14. ROS Content

A DCFH-DA assay kit was used to detect intracellular ROS levels in PC12 cells. In brief, PC12 cells were seeded into 6- and 24-well plates, pretreated with different concentrations of LICP009-3F-1a (0–500 μg/mL) for 24 h, and then treated with 400 μM CoCl_2_. After 24 h, the cells in 6-well plates were washed with PBS and incubated with 10 µM DCFH-DA for 0.5 h at 37 °C in the incubator. ROS levels were determined with flow cytometry (FCM). The cells in the 24-well plate were also incubated with 10 µM DCFH-DA for 0.5 h at 37 °C in the incubator. ROS levels were subsequently examined using the FV1200 Biological Confocal Laser Scanning Microscope (Olympus, Tokyo, Japan) at 488 nm.

### 2.15. Cell Apoptosis

PC12 cells (1 × 10^4^ cells/well) were plated onto 6- and 24-well plates with various concentrations of LICP009-3F-1a (0–500 μg/mL). After pretreatment in a CO_2_ incubator with humidified air at 37 °C for 24 h, the cells were exposed to 400 μM CoCl_2_. After incubation for 24 h, the cells in the 6-well plate were harvested and incubated with PI and Annexin V-FITC (5 μL) for 30 min at 37 °C in the dark. Cell apoptosis was examined using flow cytometry (FCM). The cells in 6-well plates were harvested, and ATP content was detected by an ATP Detection Assay Kit. The cells in the 24-well plate were incubated with MitoTracker Green (50 nM final concentration) for 30 min at 37 °C in the dark.

### 2.16. RT–PCR

The relative expression levels of Bax, Bcl2, CAT, Casp3, Gpx1, HIF-1α, SOD1, and VEGF in PC12 cells were analysed using real-time polymerase chain reaction (RT-PCR). PC12 cells (1 × 10^5^ cells/well) were plated onto 6-well plates, with three replicates for each group. After treatment with tyrisin, the cells were washed with PBS and centrifuged at 1000 rpm for 3 min at 4 °C. The resulting pellets were used for RNA extraction and subsequent RT-PCR analysis to quantify gene expression levels. Total RNA was extracted by the MiniBEST Universal RNA Extraction Kit (9767, TaKaRa, Beijing, China). RNA was converted to cDNA by the PrimeScript™ RT Reagent Kit (TaKaRa, Beijing, China). cDNA templates (Bax, Bcl2, CAT, Casp3, Gpx1, HIF-1α, SOD1, and VEGF) was used in RT-PCR, and the products were amplified using SYBR Premix Ex Taq (TaKaRa, Beijing, China) by an ABI QuantStudio 5 real-time detection system (QuantStudio^®^, Carlsbad, CA, USA). The primers are listed in Appendix A. The PCR cycling conditions were 95 °C for 15 min, followed by 40 cycles of 95 °C for 15 s, 58 °C for 30 s, and 72 °C for 32 s. The relative expression level of each gene was calculated by the 2^−ΔΔCt^ method [27].

### 2.17. Data Analysis

The data from the experiments are expressed as mean ± SD. Each experiment was performed at least in triplicate. Differences between groups were determined by one-way analysis of variance (ANOVA) using GraphPad Prism 8 software. For all results among the different groups, a *p*-value of <0.05 was regarded as statistically significant.

## 3. Results

### 3.1. Purification and Mw Analysis

The crude LICP009 polysaccharide was separated from Lycii fructus (yield 3.47%) using cascade ultrafiltration membranes of 10 kDa and 5 kDa in sequence [28]. Then, the LICP009 fraction was deproteinated, decolorized, and fractionated by DEAE-52, which was eluted with H_2_O to give LICP009-3F-1, which was further separated on a Sephadex G-50 column. After purification, LICP009-3F-1a showed a symmetrical and single peak at 18.0 min; the other small peak was identified as the solvent peak at 21.2 min. The sugar content was found to be 98.4%, while neither protein nor uronic acid was identified, indicating that it was a neutral homogeneous polysaccharide. From the HPSEC-MALLS-RID chromatogram (Figure 1A), the Mw of LICP009-3F-1a can be estimated to be 10780 Da. The monosaccharide composition was analysed by IC with PAD, and the IC traces in Figure 1B show that LICP009-3F-1a is composed of arabinose, galactose, glucose, xylose, and mannose in a ratio of 36.4:239.1:80.8:24.1:9.7. Due to the different response values of monosaccharides, Rha has an obvious chromatographic peak, but its content is very small and can be ignored.

### 3.2. FT-IR Spectroscopy

FT-IR spectroscopy was used to analyse the functional groups of LICP009-3F-1a, and the results are shown in Figure 1C. The characteristic intense stretching peak at 3426.7 cm^−1^ was assigned to the stretching vibrations of –OH. The weak absorption peaks at 2923.2 and 2892.5 cm^−1^ were due to the stretching vibration of C–H. The characteristic band near 1630.5 cm^−1^ was attributed to the bending vibrations from associated water. Generally, the typical absorption signal at approximately 1720 cm^−1^ corresponds to C=O stretching from uronic acids, and there was no absorption signal in LICP009-3F-1a, indicating that it might be a neutral polysaccharide, which is consistent with monosaccharide composition analysis [29,30]. In addition, the C-O stretching vibration and O-H deformation vibrations were observed in the regions of 1065.8 and 1415.6 cm^−1^. The weak peaks at 894.7 and 854.4 cm^−1^ confirmed the characteristic absorption of β-glycosidic bonds and α-terminal epimers [31,32].

### 3.3. X-Ray Diffractometry (XRD)

XRD is a useful tool for studying the crystallinity or amorphous nature of polysaccharides. The amounts of crystalline and amorphous regions affect some physical properties, including flexibility, solubility, viscosity, density, and functional properties, in polysaccharides; for example, amorphous polysaccharides show better solubility and water absorption than crystalline polysaccharides due to intermolecular bonds [33]. As shown in Appendix A, the XRD spectrum of LICP009-3F-1a exhibited two major peaks at 3.8 and 17.3 (2θ), and other peaks were not observed in the 10–80° 2θ region, which confirmed the presence of a semicrystalline structure in LICP009-3F-1a. In addition, the small-angle peak at 3.8 (2θ) also suggested that the polysaccharide may have an orderly pore-like structure [34].

### 3.4. Thermal Property Analysis

The thermal stability of polysaccharides is an important parameter in their applications [35]. TGA thermograms of LICP009-3F-1a are presented in Appendix A and indicate two distinct weight-loss stages. The first occurred with almost 10% mass loss below 259 °C, which suggests the presence of highly hydrophilic groups, such as absorbed water or hydrogen-bound water in the polysaccharide pores [36]. In the second stage, 36% of the initial weight was lost between 259 °C and 318 °C, indicating that the cleavage of sugar chains and structural destruction of LICP009-3F-1a occurred rapidly in a very narrow temperature range [37]. The differential scanning calorimetry (DSC) thermogram of LICP009-3F-1a exhibited one endothermic peak in the temperature range 259–318 °C, which was associated with the melting temperature of LICP009-3F-1a and was consistent with the results of TGA. These results suggest that LICP009-3F-1a shows relatively good thermal stability.

### 3.5. SEM Analysis

The biological activities of polysaccharides are closely associated with their three-dimensional structures and spatial conformations [38]. SEM is one of the main methods used to study the spatial conformation of polysaccharides [39]. Figure 1D–F present the morphological characteristics of LICP009-3F-1a. The SEM images at 500× exhibit flaky and porous structures. As the magnification increases to 2000×, we can observe that the sheet layer is accompanied by a uniform filamentous pattern. The pore diameter is approximately ten microns. The appearance of micropores is consistent with the phenomenon of finding diffraction peaks at the small corners of XRD. In addition, a large number of water molecules can be stored in the micropores, which evaporate quickly when the temperature rises, leading to the first stage of weight loss in thermogravimetric analysis.

### 3.6. Congo Red Analysis

Polysaccharide conformation refers to polysaccharides showing various forms in solution due to the differences in monosaccharide type and glycosidic bonds, mainly including random coils, rods, and triple helices [40]. The physicochemical properties and biological activities of polysaccharides are strongly influenced by their conformation. Congo red experiments are one of the main means of determining whether polysaccharides have triple-helix structures [41]. Usually, in an alkaline medium, the polysaccharide depolymerizes the triple-helix structure with increasing alkaline medium concentration, so that the maximum absorption wavelength of the complex formed with Congo red decreases [42]. The results of the Congo red experiment for LICP009-3F-1a are shown in Appendix A. NaOH was added to the complex formed between the polysaccharide and Congo red, so that the total concentration of the mixture was 0~0.5 mol/L. When the concentration was 0~0.2 mol/L, the maximum absorption wavelength of the complex formed by LICP009-3F-1a and Congo red underwent a redshift, but when the concentration of NaOH in the mixture was 0.3~0.5 mol/L, there was no obvious blueshift in the maximum absorption wavelength, indicating that the complex formed between the polysaccharide and Congo red did not change due to the depolymerization of the polysaccharide structure, in turn indicating that LICP009-3F-1a lacks a triple-helix structure.

### 3.7. Methylation Analysis

Methylation is a useful method for studying the primary structure of polysaccharides, as it can determine the glycosidic bond residues and their ratios in polysaccharides [43,44]. Methylation analysis revealed that LICP009-3F-1a contained ten different glycosidic linkages in the constituent sugars, including T-Araf, T-Glcp, T-Galp, 1,2/4-Xylp, 1,3-Galp, 1,6-Manp, 1,6-Glcp, 1,4-Glcp, 1,6-Galp, and 1,3,6-Galp (Appendix A), and the molar ratios were 8.9: 6.9: 10.2: 5.5: 5.8: 3.1: 4.8: 11.3: 16.9: 26.6. The main branching points at 1,3,6-Galp, T-Araf, T-Glcp, and T-Galp served as the terminal residues, with 1,2/4-Xylp, 1,3-Galp, 1,6-Manp, 1,6-Glcp, 1,4-Glcp, and 1,6-Galp serving as intrachain residues, indicating that LICP009-3F-12a contains a main chain and branched chains. The molar ratio between terminal units and branching points was calculated to be 0.98, which was in accordance with the fact that the number of branching points was nearly equal to the number of terminal units in the proposed structure.

### 3.8. NMR Spectroscopy

Nuclear magnetic resonance spectroscopy is an indispensable method in the structural analysis of polysaccharides [45]. Relying on NMR analysis, we can obtain the main information for polysaccharide glycosides, such as the sugar ring type, monosaccharide residue type, and glycosidic linkage site, anomeric carbon type and glycosidic linkage sequence [46,47]. The ^1^H NMR spectrum of LICP009-3F-1a is shown in Figure 2A. The signal is mainly concentrated between δ 3.0 and 5.5 ppm. Seven anomeric proton signals were observed at δ 4.37, 4.45, 4.46, 4.63, 4.67, 4.90, and 5.17 ppm, which indicated the presence of both α- and β-configurations. δ 3.2~4.0 ppm signals are the signals of H2~H6, which overlap one another and are difficult to directly identify and need to be further analysed by 2D NMR spectroscopy. By ^13^C NMR (Figure 2B), the carbon signals can primarily be observed between δ 60 and 120 ppm, with the main anomeric carbon signal peaking at δ 99.09, 102.50, 103.84, 104.69, 104.90, 105.30, and 110.62 ppm, revealing that LICP009-3F-1a has seven sugar residues. The absence of signals in the ^13^C NMR spectrum within the range of δ 170–180 suggests that LICP009-3F-1a does not contain uronic acid. Based on the spectrum of DEPT135 (Appendix A), the signal peak of -CH_2_ was revealed to be an inverted peak, indicating that the signals in this region are all -CH_2_ groups in which C6 is not substituted.

To clarify the hydrogen and carbon proton signals of each sugar residue in LICP009-3F-1a, further analysis was carried out by 2D NMR spectroscopy. Seven anomeric carbon-hydrogen proton correlation peaks could be observed from the anomeric carbon chemical shift region of the HSQC spectrum (Figure 2C). From this, combined with the methylation results, it can be determined that LICP009-3F-1a consists of seven sugar residues labelled A, B, C, D, E, F, and G. According to the peak assignment of the HSQC spectrum, the chemical shifts of the anomeric hydrogen and anomeric carbon protons of residue A were δ 5.17 ppm and δ 110.62 ppm, respectively, indicating that residue A has an α configuration. The chemical shifts of H2~H6 in residue A are assigned by the ^1^H-^1^H-COSY spectrum (Figure 2D), and the cross-peaks occur at δ 5.17/4.13 (H1/H2), δ 4.13/3.87 (H2/H3), δ 3.87/4.06 (H3/H4), δ 4.06/3.76 (H4/H5), and δ 4.06/3.76 (H5/H6). It can be inferred that the chemical shifts of H1~H6 are δ 5.17, 4.13, 3.87, 4.06, 3.76, and 3.64 ppm, respectively. Combined with the HSQC spectrum, the relative peaks of C1/H1, C2/H2, C3/H3, C4/H4, and C5/H5 indicate that the corresponding C1-C5 are δ 110.62, 82.62, 77.97, 85.22, and 62.64 ppm, respectively. Based on these findings and the literature data, residue A was determined to be α-L-Araf-(1→.

Residue B displays an anomeric proton signal at δ 4.63. The peak at δ 4.63/105.3 in the HSQC spectrum is attributed to H1/C1 in the anomeric region, suggesting that residue B has a β configuration. The ^1^H-^1^HCOSY spectrum shows cross peaks at δ 4.63/3.70, 3.70/3.8, 3.80/3.86, 3.86/3.57, 3.57/3.63, and 3.57/3.74 ppm, and the two correlation peaks between H5 and H6 indicate that the hydrogen atom is in two different chemical environments, consistent with the H2–H6 signals of residue B at δ 4.63, 3.70, 3.80, 3.86, 3.57, and 3.63/3.75 ppm. In addition, the cross peaks observed at δ 3.70/71.42, 3.80/83.10, 3.86/69.96, and 3.57/74.14 ppm correspond to H2/C2, H3/C3, H4/C4, and H5/C5, respectively. Thus, residue B was concluded to be →3)-β-D-Galp-(1→. A comparable analysis approach, incorporating HMBC and NOESY analyses, allowed for the assignment of all glycosidic bond signals, as detailed in Table 1 [48,49,50]. Notably, 5.5% of the 1,2/4-Xylp and 3.1% of the 1,6-Manp glycosyl linkages were identified through methylation; however, their corresponding NMR signals could not be detected.

The glycosidic bond signal was analysed and assigned by the HMBC spectrum (Figure 2E). The anomeric hydrogen of the glycosidic bond→6)-β-D-Galp-(1→ shows a correlation with its own C6, indicating the presence of →6)-β-D-Galp-(1→6)-β-D-Galp-(1→ (C→C). Similarly, the structure →3,6)-β-D-Galp-(1→3,6)-β-D-Galp-(1→ (D→D) can be identified in LICP009-3F-1a. The anomeric proton of →6)-β-D-Galp-(1→ is correlated to C6 of →3,6)-β-D-Galp-(1→, suggesting the presence of →6)-β-D-Galp-(1→3,6)-β-D-Galp-(1→ (C→D). Additionally, H1 of →4)-β-D-Glcp-(1→ (residue E) shows a correlation with C6 of →6)-β-D-Galp-(1→ (residue C), indicating that residue E is linked to the 6-position of residue C. The glycosidic bond →3)-β-D-Galp-(1→ exhibits correlations between H1 and C3 of →3,6)-β-D-Galp-(1→, as well as →3)-β-D-Galp-(1→ correlated with C1 and H3 of →3,6)-β-D-Galp-(1→, confirming the presence of →3)-β-D-Galp-(1→3,6)-β-D-Galp-(1→ (B→D). From the NOESY spectrum (Figure 2F), the anomeric hydrogen of →6)-α-D-Glcp-(1→ is found to correlate with H3 of →3,6)-β-D-Galp-(1→, indicating the presence of →6)-α-D-Glcp-(1→3,6)-β-D-Galp-(1→ (F→D). The anomeric hydrogen of β-D-Galp-(1→ correlates with H3 of →3)-β-D-Galp-(1→, suggesting the structure β-D-Galp-(1→3)-β-D-Galp-(1→ (G→B). Likewise, α-L-Araf-(1→ exhibits correlations at H1 and H3 of →3,6)-β-D-Galp-(1→, confirming the presence of α-L-Araf-(1→3,6)-β-D-Galp-(1→ (A→D). Based on the monosaccharide composition, infrared spectroscopy, methylation analysis, and 1D and 2D NMR spectra, the structural formula of LICP009-3F-1a indicates that its main chain consists of →4)-β-D-Glcp-(1→6)-β-D-Galp-(1→, with a branched chain of β-D-Galp-(1→3)-β-D-Galp-(1→, α-L-Araf-(1→, and→6)-α-D-Glcp-(1→ connected to the main chain through O-3 of →3,6)-β-D-Galp-(1→. The structure is shown in Figure 2G.

### 3.9. PC12 Cells Are Injured by CoCl_2_-Induced Hypoxia

The PC12 cell line is widely used in neuroscience research, such as in neuroprotection and neurotoxicity [51,52,53]. CoCl_2_ is often used as a hypoxia-mimetic agent [54]. And CoCl_2_-induced hypoxic injury has been widely used as an experimental model in in vivo and in vitro research [55,56,57]. In this study, 75, 150, 300, 450, 600, 750, and 900 μM CoCl_2_ induced hypoxic injury in PC12 cells. As shown in Figure 3A, when the concentration of CoCl_2_ reached 150 μM, cell viability had significantly decreased to 87.19% compared to that in the untreated group (*p* < 0.01). The half-maximal inhibitory concentration (IC_50_) of CoCl_2_ on PC12 cell viability was 400 μM, and this was selected as the hypoxia treatment concentration for subsequent experiments. When the concentration of CoCl_2_ reached 750 μM, CoCl_2_-induced hypoxic injury reached the maximum inhibitory concentration. Some studies have shown an IC_50_ of CoCl_2_-induced hypoxic injury to MCF-7 cells of 250 μM and an IC_50_ of CoCl_2_-induced hypoxic injury to MDA-MB-231 cells of 200 μM, and there have been differences seen among cell lines in the concentration of CoCl_2_-induced hypoxia injury [58,59].

### 3.10. LICP009-3F-1a Protects PC12 Cells Against CoCl_2_-Induced Hypoxic Injury

As shown in Figure 3B, PC12 cells were pretreated with different concentrations of LICP09-3F-1a for 24 h. After treatment with CoCl_2_, cell viability significantly decreased to 50.41% when the PC12 cells were not treated with LICP09-3F-1a. However, PC12 cells pretreated with 10, 50, 100, 200, and 500 μg/mL LICP009-3F-1a showed a significant increase in cell viability (*p* < 0.05). The results showed that LICP009-3F-1a protected PC12 cells against CoCl_2_-induced hypoxic injury. The best protective effect was observed when the concentration of LICP009-3F-1a was 100 μg/mL. Qiaoju’s studies showed that the proliferation of H9c2 cells was increased significantly by 400 μg/mL LBPs, and LICP009-3F-1a showed better protection against hypoxia [60].

### 3.11. LICP009-3F-1a Reduces ROS Levels in CoCl_2_-Induced Hypoxic Injury in PC12 Cells

Superoxide and other ROS have been recognised as harmful or toxic byproducts of aerobic metabolism and are also important signalling molecules in various physiological and pathophysiological conditions [61,62]. ROS or ROS-dependent signalling pathways are closely linked to factors associated with hypoxic environments in different ways [63,64]. We explored the effect of LICP009-3F-1a on the level of ROS in CoCl_2_-induced hypoxic injury in PC12 cells. As depicted in Figure 4, the ROS content in the hypoxia group was 1.4 times higher than that in the control group (*p* < 0.01), indicating that hypoxia led to a substantial increase in ROS levels in cells. However, the ROS levels in the experimental group treated with LICP009-3F-1a were significantly lower than those in the hypoxia group (*p* < 0.05). Different concentrations of LICP009-3F-1a revealed no obvious concentration dependence on ROS content in PC12 cells after CoCl_2_-induced hypoxic injury. When the concentration of LICP009-3F-1a was 200 μg/mL, the ROS content in cells was the lowest. As shown in Figure 5, compared with the hypoxia group, the fluorescence signal of the LICP009-3F-1a group was significantly weakened, which shows a similar trend to the flow cytometry results. Previous studies have also reported that pretreatment with LBPs could reduce the levels of ROS and protect cells and organisms against damage from excessive ROS [65,66].

### 3.12. LICP009-3F-1a Inhibits CoCl_2_-Induced Hypoxic Oxidative Stress in PC12 Cells

The reduction in ROS content in cells is caused by a combination of multiple conditions in cells, and antioxidant enzymes (SOD, CAT, and GPx) are regarded as key enzymes in the intracellular defence system against oxidative stress in cells [67,68,69]. As shown in Figure 6, LICP009-3F-1a inhibited CoCl_2_-induced oxidative stress in PC12 cells under hypoxic injury by activating the SOD, CAT, and GPx genes in PC12 cells. The activity of antioxidant enzymes was reversed and restored under LICP009-3F-1a treatment. All these results demonstrate that LICP009-3F-1a protects PC12 cells from oxidative stress induced by CoCl_2_-mediated hypoxia. Research has shown that LBPs can enhance the activities of antioxidant enzymes, such as SOD, CAT, and GPx, thereby effectively reducing ROS levels [22,70,71].

### 3.13. LICP009-3F-1a Inhibits HIF-1α and VEGF mRNA Levels in CoCl_2_-Induced Hypoxic Injury in PC12 Cells

HIF (hypoxia-inducible factor) plays a primary role in the response of cells to hypoxia and is the most important transcription factor in cells in adapting to hypoxia [72,73]. These transcription factors are sensitive to hypoxic environments, and cells adapt to hypoxic environments by modulating their expression. The VEGF gene is a key gene downstream of HIF-1α. We can see clearly from Figure 6 that the expression levels of the HIF-1α and VEGF genes were prominently increased in PC12 cells after treatment with CoCl_2_. However, the HIF-1α and VEGF genes were significantly downregulated in PC12 cells pretreated with LICP009-3F-1a (*p* < 0.01). The results demonstrate that LICP009-3F-1a protected PC12 cells against CoCl_2_-induced hypoxia injuries by suppressing the expression of the HIF-1α gene. Studies have demonstrated that LBPs can reduce the expression of HIF-1α and VEGF genes in rats subjected to hypoxia and in patients with COPD [74,75].

### 3.14. LICP009-3F-1a Inhibits CoCl_2_-Induced Hypoxia Apoptosis in PC12 Cells

Hypoxia is a significant trigger of c cell apoptosis and is closely associated with cell apoptosis [76]. The effect of LICP009-3F-1a on CoCl_2_-induced hypoxia apoptosis in PC12 cells was detected using flow cytometry. As shown in Figure 7, the apoptosis rate of PC12 cells significantly increased after CoCl_2_ induction compared to the control group (*p* < 0.01). However, LICP009-3F-1a significantly inhibited apoptosis after CoCl_2_-induced hypoxia (*p* < 0.01). To further confirm the inhibitory effect of LICP009-3F-1a on CoCl_2_-induced hypoxia apoptosis, the expression levels of apoptosis-related genes (Bcl2, Bax, and Caspase3) were measured using RT–PCR. Compared with the control group (shown in Figure 8), the mRNA levels of Bax and Caspase3 genes in PC12 cells after CoCl_2_ induction were significantly increased. In contrast, pretreatment with LICP009-3F-1a significantly inhibited Bax and Caspase3 gene expression. Otherwise, the mRNA levels of the Bcl2 gene were significantly reduced in PC12 cells. Pretreatment with LICP009-3F-1a significantly upregulated the level of the Bcl2 gene. The results show that LICP009-3F-1a can inhibit the apoptosis of PC12 cells induced by CoCl_2_-induced hypoxia. In line with findings from previous reports, LBPs downregulates the Bcl2 gene in human prostate cancer cells [77]. As shown in Figure 9, compared with the hypoxia group, the intracellular mitochondrial level and ATP level were significantly increased after the cells were pretreated with LICP009-3F-1a, suggesting that the pathway by which LICP009-3F-1a inhibits apoptosis is likely mitochondria-dependent. These results indicate that LICP009-3F-1a can inhibit the apoptosis of PC12 cells induced by CoCl_2_-induced hypoxia. Our results are inconsistent with other findings, in which LBPs effectively inhibited apoptosis in QGY7703 and SMMC-7721 cells [78,79].

## 4. Conclusions

In this study, we identified a novel neutral, water-soluble polysaccharide, LICP009-3F-1a, which was separated and purified from *Lycium barbarum* L. Preliminary homogeneity, molecular weight, monosaccharide composition, thermogravimetry, FT-IR spectroscopy, methylation, and NMR analyses were conducted to examine its structure. The results revealed that LICP009-3F-1a had a molecular mass of 10,780 Da. The main chain of the structure was →4)-β-D-Glcp-(1→6)-β-D-Galp-(1→, with a branched chain of β-D-Galp-(1→3)-β-D-Galp-(1→, α-L-Araf-(1→ and →6)-α-D-Glcp-(1→, connected to the main chain through O-3 of →3,6)-β-D-Galp-(1→. In vitro experiments revealed that LICP009-3F-1a could protect PC12 cells from CoCl_2_-induced hypoxia injury by reversing and restoring the activity of the antioxidant enzymes SOD, CAT, and GPx and downregulating HIF-1α and VEGF expression. Moreover, LICP009-3F-1a inhibited mitochondria-dependent apoptosis by lowering the expression levels of caspase3 and Bax while increasing the expression levels of Bcl-2. LICP009-3F-1a shows a significant neuroprotective effect and shows promise for use in applications in ischaemic stroke.

## Figures and Tables

**Figure 1 foods-14-00339-f001:**
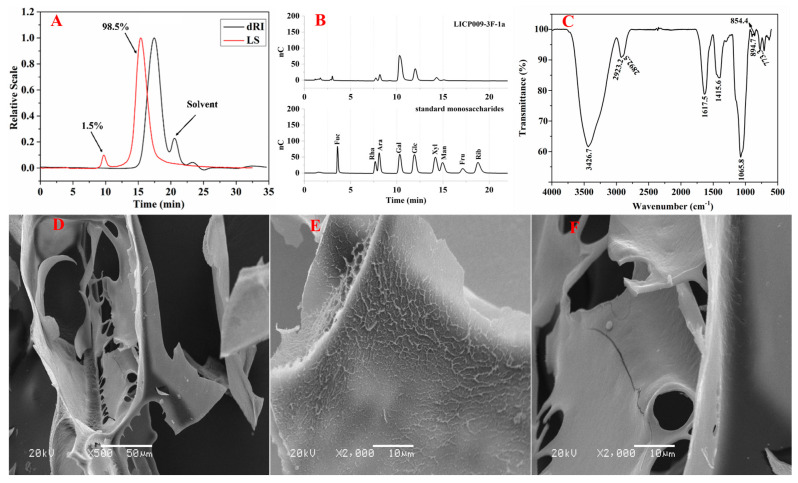
(**A**) Profile of SEC-MALLS of polysaccharide LICP009-3F-1a, (**B**) IC trace of standard monosaccharides and IC trace of LICP009-3F-1a, (**C**) FT-IR spectrum of LICP009-3F-1a, (**D**–**F**) SEM spectra of LICP009-3F-1a with magnifications of 500× and 2000×.

**Figure 2 foods-14-00339-f002:**
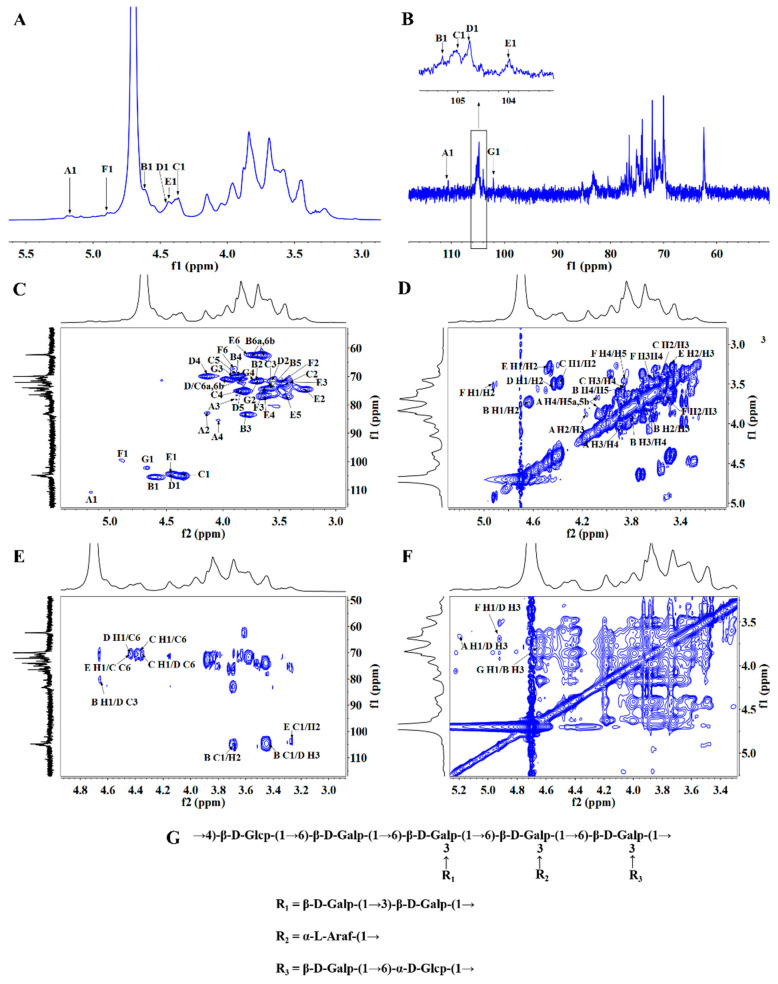
(**A**) ^1^H NMR spectra, (**B**) ^13^C NMR spectra, (**C**) HSQC spectra, (**D**) ^1^H-^1^H-COSY spectra, (**E**) HMBC spectra, (**F**) NOESY spectra, and (**G**) the proposed structure of LICP009-3F-1a.

**Figure 3 foods-14-00339-f003:**
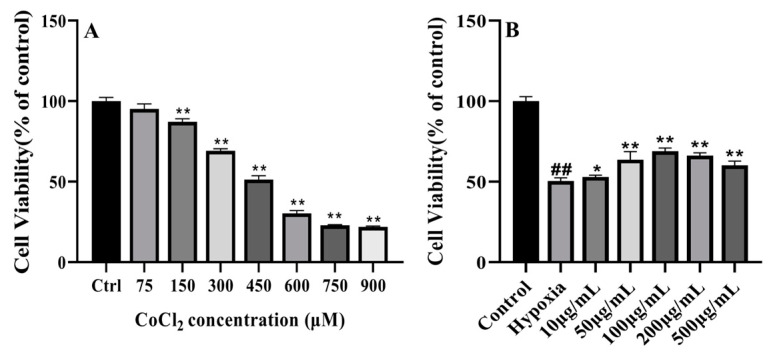
(**A**) Effects of different concentrations of CoCl_2_ (75, 150, 300, 450, 600, 750, and 900 μM) on PC12 cell proliferation. (**B**) Proliferation of PC12 cells treated with CoCl_2_ in the presence of different concentrations of LICP009-3F-1a (10, 50, 100, 200, and 500 μg/mL). The results are expressed as the means ± SD, *n* = 5. ^##^
*p* < 0.01 vs. the control group, * *p* < 0.05, ** *p* < 0.01 vs. the hypoxia group.

**Figure 4 foods-14-00339-f004:**
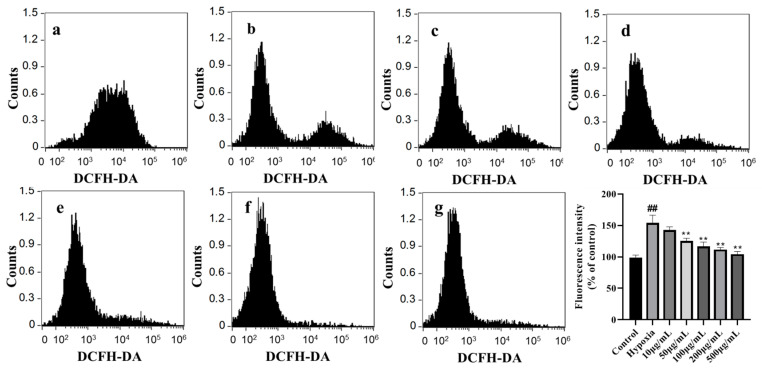
Effects of different concentrations (10, 50, 100, 200, and 500 μg/mL) of LICP009-3F-1a on ROS levels in PC12 cells treated with CoCl_2_. (**a**) The control group, (**b**) the hypoxia group, (**c**) the 10 μg/mL LICP009-3F-1a treatment group, (**d**) the 50 μg/mL LICP009-3F-1a treatment group, (**e**) the 100 μg/mL LICP009-3F-1a treatment group, (**f**) the 200 μg/mL LICP009-3F-1a treatment group, (**g**) the 500 μg/mL LICP009-3F-1a treatment group. The results are expressed as the means ± SD, n = 3. ^##^
*p* < 0.01 vs. the control group, ** *p* < 0.01 vs. the hypoxia group.

**Figure 5 foods-14-00339-f005:**
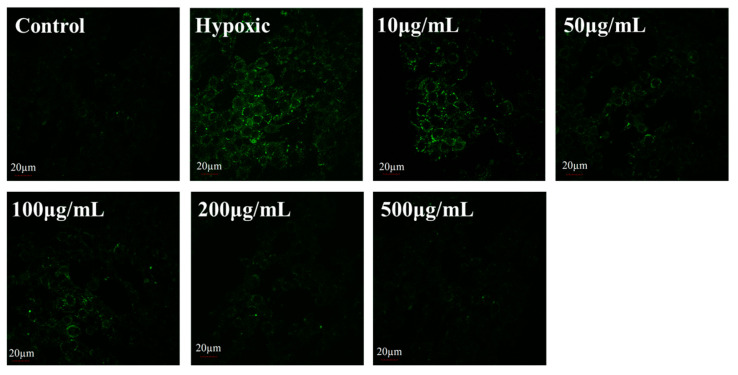
Fluorescence images of CoCl_2_-induced ROS detected in PC12 cells treated with LICP009-3F-1a (0, 50, 100, 200, and 500 μg/mL).

**Figure 6 foods-14-00339-f006:**
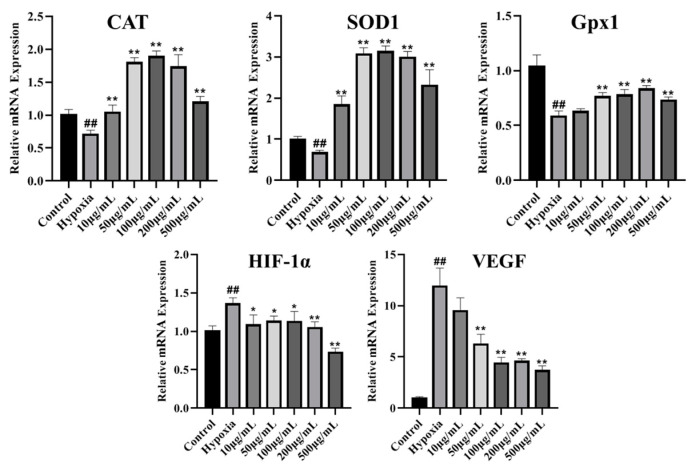
Effects of different concentrations of LICP009-3F-1a (10, 50, 100, 200, and 500 μg/mL) on the mRNA levels of CAT, SOD1, Gpx1, HIF-1α, and VEGF in PC12 cells induced by CoCl_2_. The results are expressed as the means ± SD, n = 3, ^##^
*p* < 0.01 vs. the control group, * *p* < 0.05, ** *p* < 0.01 vs. the hypoxia group.

**Figure 7 foods-14-00339-f007:**
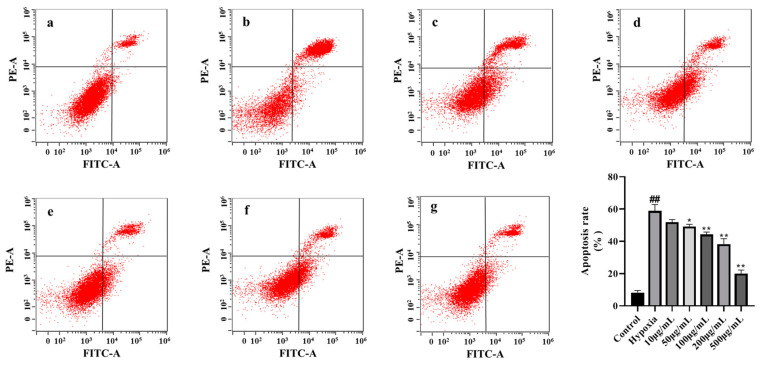
Effects of different concentrations (10, 50, 100, 200, and 500 μg/mL) of LICP009-3F-1a on CoCl_2_-induced apoptosis in PC12 cells. (**a**) The control group, (**b**) the model group, (**c**) the 10 μg/mL LICP009-3F-1a treatment group, (**d**) the 50 μg/mL LICP009-3F-1a treatment group, (**e**) the 100 μg/mL LICP009-3F-1a treatment group, (**f**) the 200 μg/mL LICP009-3F-1a treatment group, (**g**) the 500 μg/mL LICP009-3F-1a treatment group. The results are expressed as the means ± SD, n = 3. ^##^ *p* < 0.01 vs. the control group, * *p* < 0.05, ** *p* < 0.01 vs. the hypoxia group.

**Figure 8 foods-14-00339-f008:**
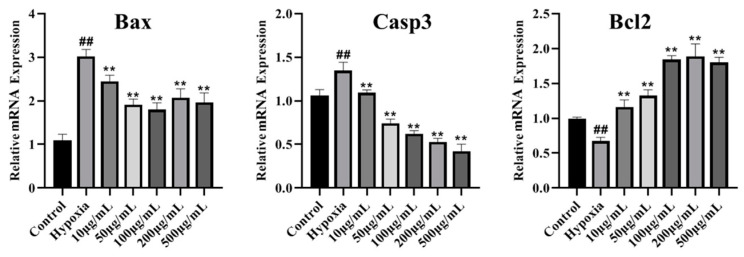
Effects of different concentrations (10, 50, 100, 200, and 500 μg/mL) of LICP009-3F-1a on the mRNA levels of Bax, Bcl2, and Casp3 in PC12 cells treated with CoCl_2_. The results are expressed as the means ± SD, n = 3. ^##^
*p* < 0.01 vs. the control group, ** *p* < 0.01 vs. the hypoxia group.

**Figure 9 foods-14-00339-f009:**
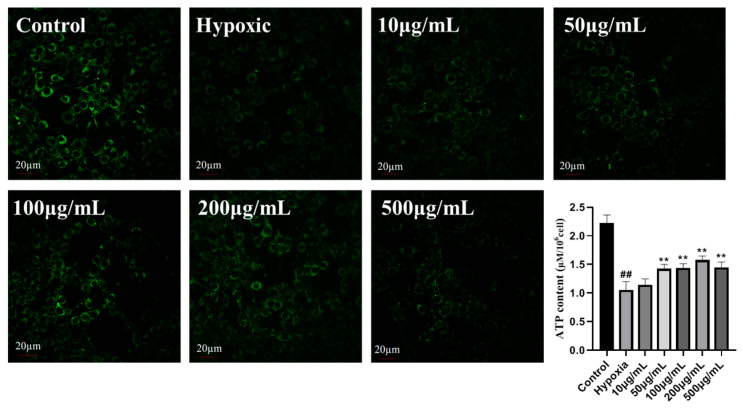
Effects of different concentrations (10, 50, 100, 200, and 500 μg/mL) of LICP009-3F-1a on the levels of mitochondria and ATP in PC12 cells treated with CoCl_2_. The results are expressed as the means ± SD, n = 3. ^##^ *p* < 0.01 vs. the control group, ** *p* < 0.01 vs. the hypoxia group.

**Table 1 foods-14-00339-t001:** ^1^H and ^13^C NMR spectral assignments for LICP009-3F-1a (ppm).

Glycosyl Residues	H1/C1	H2/C2	H3/C3	H4/C4	H5/C5	H6a, H5b/C6	H6b
(A) α-L-Ara*f*-(1→	5.17	4.13	3.87	4.06	3.76	3.64	
	110.62	82.62	77.97	85.22	62.64		
(B) →3)-β-D-Gal*p*-(1→	4.63	3.70	3.80	3.86	3.57	3.63	3.74
	105.3	71.42	83.1	69.96	74.14	62.45	
(C) →6)-β-D-Gal*p*-(1→	4.37	3.44	3.58	3.86	3.88	3.95	3.83
	104.9	72.16	73.93	74.96	69.87	70.50	
(D) →3,6)-β-D-Gal*p*-(1→	4.45	3.57	3.63	4.16	3.84	3.96	3.86
	104.69	71.31	81.5	69.82	74.81	70.76	
(E) →4)-β-D-Glc*p*-(1→	4.46	3.28	3.43	3.64	3.47	3.95	3.79
	103.84	74.17	71.58	77.80	76.69	61.67	
(F) →6)-α-D-Glc*p*-(1→	4.90	3.50	3.65	3.45	3.84	3.90	
	99.09	72.73	74.63	70.87	71.47	66.97	
(G) β-D-Gal*p*-(1→	4.67	3.70	3.90	3.75	3.63	3.74	3.62
	102.5	71.5	70.05	71.71	76.4	62.45	

## Data Availability

The original contributions presented in the study are included in the article/Appendix A, further inquiries can be directed to the corresponding author.

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
