# Peer review of "Structural Characterization of, and Protective Effects Against, CoCl2-Induced Hypoxia Injury to a Novel Neutral Polysaccharide from Lycium barbarum L."

_foods, 2025, doi:10.3390/foods14030339_

Round 1
Reviewer 1 Report
Comments and Suggestions for Authors
All scientific names should be in Italics.
The Material and methods section should be supported by citations or clearly mentioned that methods were developed in the corresponding labs.
Author Response
Comments 1: [All scientific names should be in Italics.]
Response 1: Thank you for pointing this out. We agree with this comment. Therefore, we checked all scientific names in the manuscript, and changed them into Italics.
Comments 2: [The Material and methods section should be supported by citations or clearly mentioned that methods were developed in the corresponding labs.]
Response 2: Thank you for pointing this out. We agree with this comment. Therefore, We checked the Material and methods section, and clearly mentioned that the methods were developed in the corresponding labs.

Reviewer 2 Report
Comments and Suggestions for Authors
The manuscript shows a 40% similarity with other reports, some of them yours. The limit accepted for this is no more than 20%; please decrease the similarity with other documents.
Abstract: What do you mean with "a single oxidative stress"?
Please improve the quality of Figure 6 and 7.
I suggest you to adjust the title, because you also evaluate the effect of your polysaccharide on cell apoptosis not only against oxidative stress.
In the introduction add information about the effect of functional polysaccharides on cell apoptosis.
Author Response
Comments 1: [The manuscript shows a 40% similarity with other reports, some of them yours. The limit accepted for this is no more than 20%; please decrease the similarity with other documents.]
Response 1: Thank you for pointing this out. We agree with this comment. Therefore, We have modified the manuscript to decrease the similarity with other documents.
Comments 2: [Abstract: What do you mean with "a single oxidative stress"?]
Response 2: Thank you for pointing this out. We agree with this comment. The meaning of “A single Oxidative stress” is that oxidative stress plays a significant role in ischemic stroke. And the expression of “A single Oxidative stress” isn’t good, so we changed it into “Oxidative stress is related to the occurrence and development of ischaemic stroke closely.“
Comments 3: [Please improve the quality of Figure 6 and 7.]
Response 3: Thank you for pointing this out. We agree with this comment. We have improve the quality of Figure 6 and 7.
Comments 4: [I suggest you to adjust the title, because you also evaluate the effect of your polysaccharide on cell apoptosis not only against oxidative stress. In the introduction add information about the effect of functional polysaccharides on cell apoptosis.]
Response 4: Thank you for pointing this out. We agree with this comment. We have added information about the effect of functional polysaccharides on cell apoptosis in the introduction, the revised manuscript this change can be found in line 58-59. However, we didn't change the title. In the title, the emphasis is on the protective effects of our polysaccharide against CoCl2-induced hypoxia injury. This not only means the effect of our polysaccharide on combating oxidative stress but also its effect on preventing cell apoptosis.

Reviewer 3 Report
Comments and Suggestions for Authors
In line 34, add the means ROS like in line 45
The FTIR measurement was done in pellet or by ATR, specify
Homogenize throughout the text if abbreviations such as NaOH are to be used or put the full name as in line 120.
In line 122 in GC-MS, what column was used? Was it an ionic tramp? Temperature tramp and injector
The methodology describes how its polysaccharide was obtained, as described in reference 24, and the results mention that a cascade was made according to reference 28. Which reference is correct?
In line 197, what means Rha?
Line 261 mentions Figure S3, but the figure is not found.
Figure 2 could be in the corresponding text after the NMR results.
Figure 9 could be in the corresponding text before the conclusion.
It has 36% plagiarism and agrees with the highest percentage of an article published by the authors, so it is suggested to reduce the number, look for other sources, and compare the results obtained with other sources.
I think abbreviations should be considered. Explain what they mean at the beginning, and then use them.

Author Response
Comments 1: In line 34, add the means ROS like in line 45]
Response 1: Thank you for pointing this out. We agree with this comment. Therefore, We have added information about ROS, the revised manuscript this change can be found in line 34 and line 45.
Comments 2: [The FTIR measurement was done in pellet or by ATR, specify]
Response 2: Thank you for pointing this out. We agree with this comment. The FTIR measurement was done by ATR, The ATR method is widely used for infrared analysis of polysaccharides.
Comments 3: [Homogenize throughout the text if Homogenize throughout the text if abbreviations such as NaOH are to be used or put the full name as in line 120.
Response 3: Thank you for pointing this out. We agree with this comment. We have checked Homogenization throughout the text such as NaOH in our revised manuscript.
Comments 4: [In line 122 in GC-MS, what column was used? Was it an ionic tramp? Temperature tramp and injector]
Response 4: Thank you for pointing this out. We agree with this comment. The chromatographic system used is an Agilent gas chromatography system (Agilent 7890A; Agilent Technologies, USA) with a BPX70 column (30 m × 0.25 mm × 0.25 µm, SGE, Australia). The injection volume was 1 µl, with a split ratio of 10:1. The carrier gas was high-purity helium at a flow rate of 1.5 ml/min. The initial oven temperature was 140°C, held for 2 minutes, then increased at a rate of 3°C/min to 230°C, where it was held for 3 minutes. The analytical instrument used in this experiment was an Agilent 7890A-5977B gas chromatography-mass spectrometry (GC-MS) system (Agilent Technologies Inc., CA, USA), with an autosampler model G4567A.
Comments 5: [The methodology describes how its polysaccharide was obtained, as described in reference 24, and the results mention that a cascade was made according to reference 28. Which reference is correct? ]
Response 5: Thank you for pointing this out. The methodology describes how the polysaccharide was obtained, as described in reference 24, and it was our previous procedure for obtaining our polysaccharide. The results mention that a cascade was made according to reference 28. The reference we mentioned is to emphasize the ultrafiltration membranes we used. Both of them are correct.
Comments 6: [In line 197, what means Rha?]
Response 6: Thank you for pointing this out. “Rha” is the abbreviation for Rhamnose.
Comments 7: [Line 261 mentions Figure S3, but the figure is not found.]
Response 7: Thank you for pointing this out. We agree with this comment. We have added Figure S3 in foods-3318343-supplementary.
Comments 8: [Figure 2 could be in the corresponding text after the NMR results. Figure 9 could be in the corresponding text before the conclusion.]
Response 8: Thank you for pointing this out. We agree with this comment. We have modified the position of the picture.
Comments 9: [It has 36% plagiarism and agrees with the highest percentage of an article published by the authors, so it is suggested to reduce the number, look for other sources, and compare the results obtained with other sources. ]
Response 9: Thank you for pointing this out. We agree with this comment. We have modified the manuscript to decrease the similarity with other documents.
Comments 10: [ I think abbreviations should be considered. Explain what they mean at the beginning, and then use them. ]
Response 10: Thank you for pointing this out. We agree with this comment. We have added the full names and abbreviations of scientific terms at the beginning, and then used them.
